



# Numerical issues of the Total Exchange Flow (TEF) analysis framework for quantifying estuarine circulation

Marvin Lorenz[1], Knut Klingbeil[1], Parker MacCready[2], and Hans Burchard[1]

[1]Leibniz Institute for Baltic Sea Research Warnemünde, Rostock, Germany
[2]University of Washington, Seattle, Washington

**Correspondence:** Marvin Lorenz (marvin.lorenz@io-warnemuende.de)

**Abstract.** For more than a century, estuarine exchange flow has been quantified by means of the Knudsen relations which connect bulk quantities such as inflow and outflow volume fluxes and salinities. These relations are closely linked to estuarine mixing. The recently developed Total Exchange flow (TEF) which uses salinity coordinates to calculate these bulk quantities allows an exact formulation of the Knudsen relations in realistic cases. There are however numerical issues, since the original

5  method does not converge to the TEF bulk values for an increasing number of salinity classes. In the present study, this problem is investigated and the method of *dividing salinities*, described by MacCready et al. (2018), is mathematically introduced. A challenging yet compact analytical scenario for a well-mixed estuarine exchange flow is investigated for both methods, showing the proper convergence of the dividing salinity method. Furthermore, the dividing salinity method is applied to model results of the Baltic Sea to demonstrate the analysis of realistic exchange flows and exchange flows with more than two layers.

## 1  Introduction

The *Total Exchange Flow* (TEF) analysis framework calculates time-averaged net volume and mass transports between enclosed volumes of the ocean and ambient water masses, sorted by salinity classes. Since inflow and outflow occurring at the same salinity compensate, TEF characterises the net exchange flow with the ambient ocean. Salinity rather than density or temperature is used as a coordinate for calculating estuarine exchange flow, since only the salt budget is entirely controlled

15  by the exchange flow. Therefore, salt is the only conserved quantity. In contrast, temperature and thus density are additionally affected by the freshwater run-off and the surface heat fluxes.

A first bulk approach based on inflow and outflow salinity and volume transport had been developed and applied to the exchange flow of the Baltic Sea by Knudsen (1900). The theoretical framework based on a continuous salinity space had first been developed by Walin (1977), and later been applied to exchange flow in the Baltic Sea (Walin, 1981). A comparable

20  framework had been applied by Döös and Webb (1994) for quantifying meridional overturning circulation in the Southern Ocean. Both the bulk concept by Knudsen (1900) and the continuous concept by Walin (1977) had been consistently combined by MacCready (2011) who also coined the term *Total Exchange Flow*, TEF.





TEF considers a time-averaged transport of a tracer $c$, $Q^c$, through the cross-sectional area $A(S)$, which has a salinity $s$ above a specific value $S$. $Q^c$ is defined as

$$Q^c(S) = \left\langle \int_{A(S)} c\,u\,\mathrm{d}A \right\rangle, \tag{1}$$

where $u$ is the incoming velocity normal to $A(S)$ with the definition that positive $u$ brings water into the estuary and $\langle\rangle$ denotes temporal averaging. The exchange profile of tracer flux per salinity as functions of the salinity is then obtained by differentiating $Q^c(S)$ with respect to $S$:

$$q^c(S) = -\frac{\partial Q^c(S)}{\partial S}, \tag{2}$$

MacCready (2011) calculates the inflowing and outflowing bulk fluxes by integrating over positive and negative parts of $q^c$:

$$Q^c_{\mathrm{in}} = \int_{S_{\mathrm{min}}}^{S_{\mathrm{max}}} (q^c)^+ \,\mathrm{d}S, \quad Q^c_{\mathrm{out}} = \int_{S_{\mathrm{min}}}^{S_{\mathrm{max}}} (q^c)^- \,\mathrm{d}S, \tag{3}$$

where for any function $a$, the positive part is calculated as $(a)^+ = \max(a, 0)$ and the negative part is calculated as $(a)^- = \min(a, 0)$. In (3), $S_{\mathrm{min}}$ and $S_{\mathrm{max}}$ are the minimum and maximum salinities. We will call this method of integrating positive and negative contributions separately to obtain $Q^c_{\mathrm{in}}$ and $Q^c_{\mathrm{out}}$ *sign method* in the following. Bulk salinities are defined as the fractions between the salinity fluxes, $Q^s_{\mathrm{in}}$ and $Q^s_{\mathrm{out}}$, and the volume fluxes, $Q_{\mathrm{in}} = Q^1_{\mathrm{in}}$ and $Q_{\mathrm{out}} = Q^1_{\mathrm{out}}$:

$$s_{\mathrm{in}} = \frac{Q^s_{\mathrm{in}}}{Q_{\mathrm{in}}}, \quad s_{\mathrm{out}} = \frac{Q^s_{\mathrm{out}}}{Q_{\mathrm{out}}}. \tag{4}$$

Recently, Klingbeil et al. (2018a) showed the relation between TEF and Thickness Weighted Averaging. The concepts by Knudsen (1900), Walin (1977) and MacCready (2011) were focussed on estuarine systems, which are characterised by distinct volume inflow $Q_r$ of water masses of (almost) zero salinity. The exchange flow between the estuary and the ocean is described by the Knudsen bulk values, which are volume inflow and outflow of saline water masses, $Q_{\mathrm{in}}$ and $Q_{\mathrm{out}}$ as well as associated inflow and outflow salinities, $s_{\mathrm{in}}$ and $s_{\mathrm{out}}$. The Total Exchange Flow provides one consistent calculation method for these bulk values, which for this case describe the net exchange flow. Since there is no clear definition of the Knudsen bulk values, we will call these Total Exchange Flow *TEF bulk values* to distinguish between other bulk values which also fulfill the Knudsen relations, e.g. bulk values computed from Eulerian version of TEF. The Knudsen relations have been reviewed in detail for exchange flow in the Western Baltic Sea by Burchard et al. (2018a). Recently, MacCready et al. (2018) showed how the bulk concept can be used to estimate the volume-integrated average mixing $M$ (defined as the rate of reduction of the net salinity variance due to mixing) in estuaries: $M \approx s_{\mathrm{in}} s_{\mathrm{out}} Q_r$, i.e. the volume-integrated average mixing in an estuary is approximated by the product of inflow and outflow salinity with the estuarine freshwater supply. This mixing estimate by MacCready et al. (2018) approximates the TEF-based exact formulations developed by Burchard et al. (2018b).

Since the TEF analysis framework is continuous in salinity, a discretisation in salinity space is required when analysing results from numerical model simulations or field observations. In their Appendix A2, Klingbeil et al. (2018b) presented the





remapping of discrete data into bins. As a result, the output of a numerical model consists of a finite number of transport values associated with the same number of discrete salinities. TEF profiles computed from numerical model output can be noisy, i.e. sign changes in $q^c$, when the number of discrete salinity classes $N$ is chosen too high as discussed by MacCready et al. (2018). This leads to incorrect TEF bulk values since as described above only the sign is used to distinguish between inflow

and outflow. In the limit of $N \to \infty$, meaning each transport value has its own salinity class, the bulk values do not converge towards the correct ones, but rather towards absolute values, e.g. for the volume inflow:

$$\lim_{N \to \infty} Q_{in}^{\mathrm{sign}}(N) = Q_{in}^{\mathrm{abs}} \neq Q_{\mathrm{in}}, \tag{5}$$

with $Q_{in}^{\mathrm{sign}}$ being the inflowing volume flux computed with the sign method, defined in (3) with $c = 1$, and

$$Q_{\mathrm{in}}^{\mathrm{abs}} = \left\langle \int_A u^+ \mathrm{d}A \right\rangle. \tag{6}$$

MacCready et al. (2018) suggested a way around this problem by finding a *dividing salinity* $S_{\mathrm{div}}$ which separates the inflow and outflow of a classical two-layer estuary with inflow at high and outflow at low salinity classes. The bulk values for inflow and outflow are then obtained by integrating:

$$Q_{\mathrm{in}}^c = \int_{S_{\mathrm{div}}}^{S_{\mathrm{max}}} q^c \, \mathrm{d}S = \max(Q^c(S)), \quad Q_{\mathrm{out}}^c = \int_{S_{\mathrm{min}}}^{S_{\mathrm{div}}} q^c \, \mathrm{d}S. \tag{7}$$

It should be noted that analytically and for smooth $q^c$ with only one zero crossing both methods coincide. We will show in

Sect. 2 the different convergence behaviours and will show that the *dividing salinity method* indeed converges towards robust TEF bulk values, e.g. for in inflowing volume flux:

$$\lim_{N \to \infty} Q_{\mathrm{in}}^{\mathrm{div}}(N) = Q_{\mathrm{in}}, \tag{8}$$

where $Q_{\mathrm{in}}^{\mathrm{div}}$ denotes the infowing volume flux computed with the dividing salinity method (7) for $c = 1$.

Obviously, this dividing salinity method only works for classical exchange flows. In Section 3 we will introduce an extended

formulation of the dividing salinity method which includes inverse estuaries (outflow at high salinities and inflow at low salinities) as well as exchange flows with more than two exchange layers in salinity space. Furthermore, in Section 3.2 the corresponding discrete description is presented. Afterwards in Section 4, the extended method is applied to numerical output from a model of the Baltic Sea, before we conclude in Section 5.

## 2 Convergence analysis for an analytical classical exchange flow

To demonstrate the different convergence behaviours of the sign method and the dividing salinity method, we take the analytical example from Burchard et al. (2018b). It describes a well-mixed tidal flow with oscillating salinity as it occurs e.g. in the Wadden Sea (Purkiani et al., 2015). The velocity and salinity are given by

$$u(t) = u_r + u_a \cos(\omega t); \quad s(t) = s_r + s_a \cos(\omega t + \phi), \tag{9}$$





with the residual velocity $u_r < 0$, the residual salinity $s_r$, the velocity and salinity amplitudes $u_a > 0$ and $s_a > 0$ with $s_r - s_a \geq$ 0, the tidal frequency $\omega = 2\pi/T$ with the tidal period $T$, and the tidal phase $\phi$. The tidally averaged salinity transport is given by

$$\frac{1}{T} \int_0^T u s \, \mathrm{d}t = u_r s_r + \frac{u_a s_a}{2} \cos(\phi). \tag{10}$$

Zero residual salt transport therefore requires

$$\cos(\phi) = -2 \frac{u_r s_r}{u_a s_a} \quad \text{with} \quad u_a s_a \geq 2|u_r| s_r. \tag{11}$$

Fig. 1 shows an example for $u(t)$, $s(t)$ and $u(t) \cdot s(t)$ with $A = 10000$ m$^2$, $u_r = -0.1$ m s$^{-1}$, $u_a = 1$ m s$^{-1}$, $s_r = 20$ g/kg and $s_a = 10$ g/kg resulting in $\phi = -1.16 = -0.185 \cdot 2\pi$. $Q(S)$, $Q^s(S)$ and $S_{\mathrm{div}}$ can be calculated analytically, see Appendix A. By means of (3) or (7), which coincide by definition for this case, and (4), the inflow and outflow volume fluxes and salinities,

$Q_{\mathrm{in}}$, $Q_{\mathrm{out}}$, $s_{\mathrm{in}}$, and $s_{\mathrm{out}}$, can then be exactly calculated. The resulting analytical TEF bulk values are $Q_{\mathrm{in}} = 813.240$ m$^3$s$^{-1}$, $Q_{\mathrm{out}} = -1813.240$ m$^3$s$^{-1}$, $s_{\mathrm{in}} = 28.424$ g/kg, and $s_{\mathrm{out}} = 12.748$ g/kg. The analytical profiles for $Q(S)$ and $q(S)$ are shown Fig. 2d.

     To visualise why only the dividing salinity method is converging towards the real bulk values, we created a time series of $I = 10^5$ time steps of (9) and computed $q(S)$ and $q^s(S)$ for varying number $N$ of salinity classes between $S_{\mathrm{min}} = 10$ g/kg

and $S_{\mathrm{max}} = 31$ g/kg, see Fig. 2. In Fig. 2a ($N = 128$) the small $N$ leads to smooth profiles for both $q$ and $Q$. Profiles of higher numbers ($N = 1024$, $N = 8092$) of salinity classes exhibit more noisy $q$ but apparently still smooth $Q$ (Fig. 2b,c). Comparison with the analytical solution (Fig. 2d) shows $Q$ is similar for all $N$ and $q$ becomes more noisy. This is a result of the numerical discretisation of the data. Most likely, the numerical values (e.g. due to round off errors) for these salinities are all different. Other than in continuous salinity space where inflows and outflows in the same salinity class partially compensate, the

corresponding discrete values could be associated with different salinity classes and the compensation does not occur anymore, resulting in noisy profiles. $Q$ only appears to be smooth, but the noise is of course apparent since $Q$ and $q$ are dependent of each other as from (2) follows

$$Q^c(S) = - \int_{S' > S} q^c(S') \, \mathrm{d}S'. \tag{12}$$

The differences of the noises are because of the scaling with the salinity bin size $\delta S$ which decreases for increasing $N$.

To study the convergence of the two different methods, the sign and dividing salinity method, one can compare the errors in discrete form to the analytical values. Fig. 3 shows the relative error , $|Q_{\mathrm{in}}(I, N) - Q_{\mathrm{in}}|/|Q_{\mathrm{in}}|$, of the numerically computed inflow bulk values in dependency of the number of time steps $I$ and the number of salinity classes $N$. For this analytical scenario both methods coincide for a small number of salinity classes. For constant $I$ and increasing $N$ the error of the sign method increases beyond a critical number of salinity classes and converges to the error of the absolute values (black line, see

(6)), whereas the dividing salinity method converges towards a small constant relative error. The critical point where the error of the sign method increases is different for each number of time steps. We did not find any general relation of this critical point





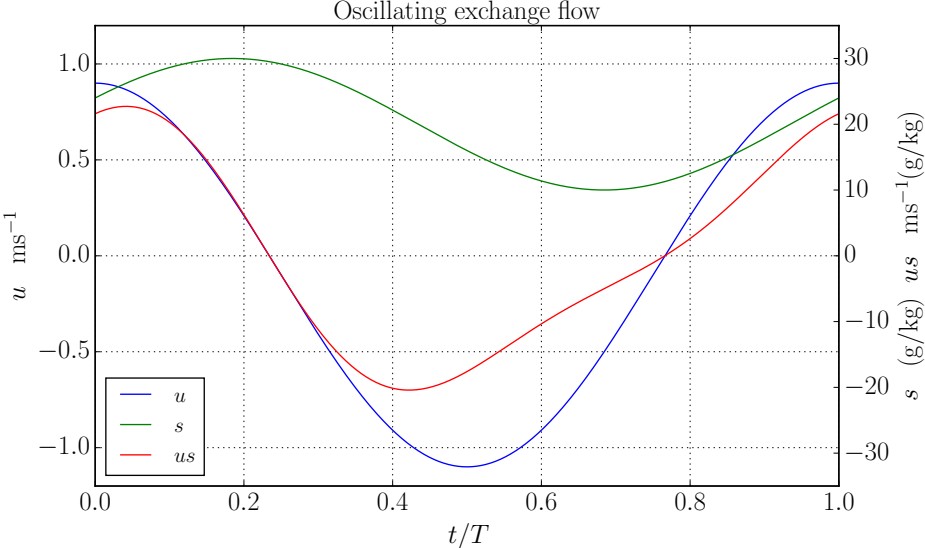

**Figure 1.** Velocity (blue), salinity (green) and salinity flux (red) time series for the oscillating exchange flow scenario (9).

to $I/N$. The error of the dividing salinity method decreases continuously with an increasing number of time steps $I$, showing that indeed the dividing salinity method converges towards the correct bulk values. Interestingly, there is almost no difference for $I = 10^3$ and $I = 10^4$, and $I = 10^5$ and $I = 10^6$ for the dividing salinity method which is due to the compensation of the added values in the data of $u(t)$ and $s(t)$. For this scenario of a well-mixed estuary one needs $I \geq 1000$ time steps per tidal period $T$, meaning one data point every minute or less, to find the transports $Q_{in}$ with an error less than 0.1% with the dividing salinity method. This is due the strong time dependency of the problem. For a stationary problem one point in time would be sufficient to find the correct exchange flow.

## 3 Extended dividing salinity method

### 3.1 Mathematical formulation

Encouraged by the good convergence behaviour of the dividing salinity method demonstrated in the previous section, we introduce here a general formulation which includes inverse estuaries and exchange flows with more than two layers. The general idea is to identify the salinities which divide $q^c$ into inflowing and outflowing parts. This corresponds to zero crossings, dividing $q^c > 0$ and $q^c < 0$. Analytically the zero crossings are calculated by solving $q^c(S_{div}) = 0$ for $S_{div}$. But as the discrete $q^c$ might be very noisy with too many zero crossings, see Sect. 2, we propose finding the extrema of the discrete $Q^c$ profiles, which share the same salinities as the zero crossings. Fig. 4 shows a hypothetical exchange flows of four layers, separated by five dividing salinities which can be sorted in ascending order: $S_{min} = S_{div,1} < S_{div,2} < S_{div,3} < S_{div,4} < S_{div,5} = S_{max}$. The





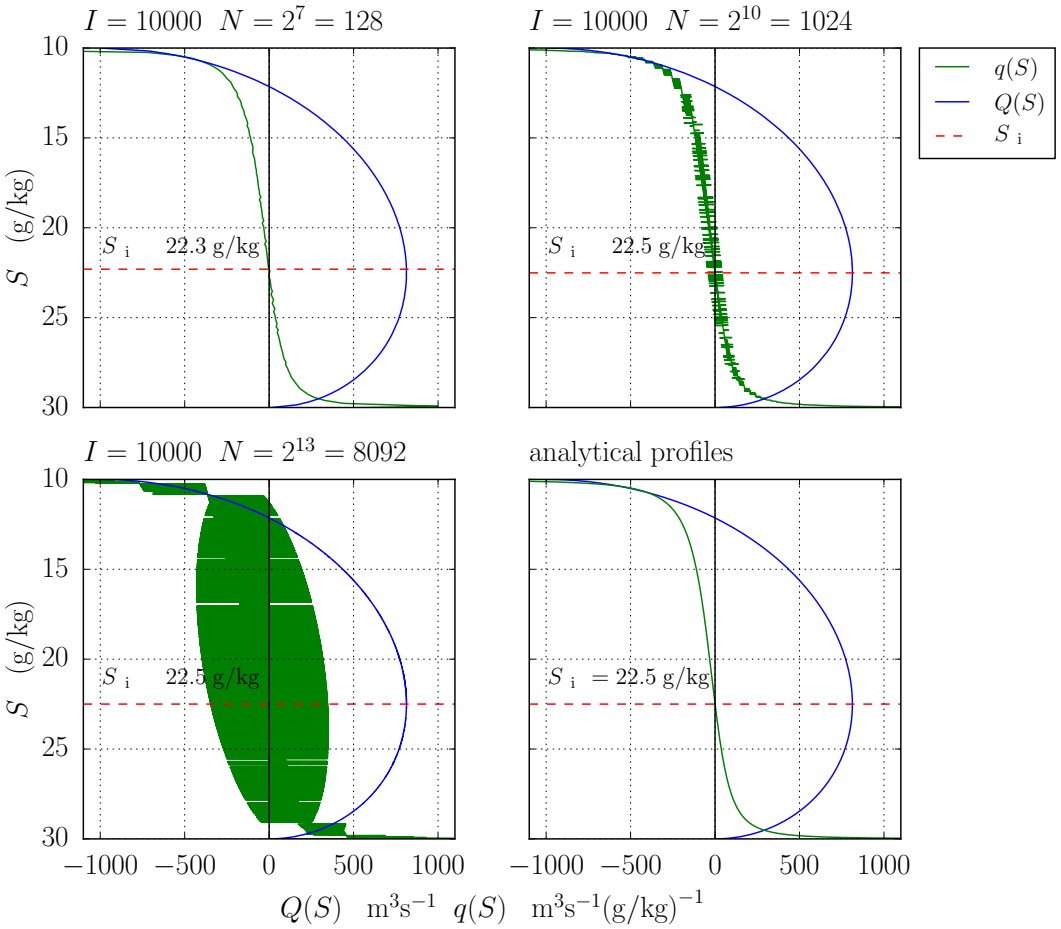

**Figure 2.** Numerically, a)-c), and analytically, d), found $Q(S)$ (blue), $q(S)$ (green), dividing salinity, $S_{\text{div}}$, (dashed, red) for the oscillating exchange flow scenario (Sect. 2) for $I = 10^4$ time steps for one tidal cycle and varying number of salinity classes $N$. With increasing $N$ the $q$ becomes more noisy, whereas $Q$ seems unchanged.





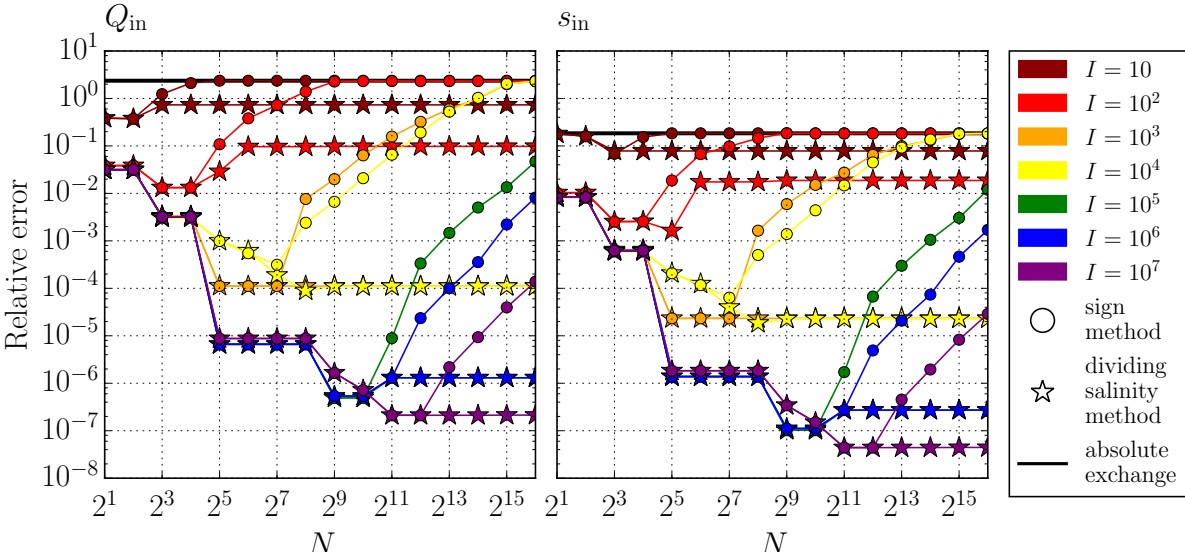

**Figure 3.** Comparison of the relative error of numerically calculated bulk values to the analytical values, a) $Q_{\text{in}}$ and b) $s_{\text{in}}$ in dependency of the number of time steps $I$ (color) and salinity classes $N$. The sign method (dots), (3) (MacCready, 2011) and the dividing salinity method (stars), (7) (MacCready et al., 2018) coincide for a small number of salinity classes, but the error of the sign method converges in the limit of large $N$ towards the the error of the absolute bulk values (black line), defined in (6). The errors of both methods decrease with increasing number of time steps $I$.

fluxes $\Delta Q_j^c$ in each layer can be calculated by

$$\Delta Q_j^c = \int_{S_{\text{div},j}}^{S_{\text{div},j+1}} q^c \, \mathrm{d}S = Q^c(S_{\text{div},j+1}) - Q^c(S_{\text{div},j}). \tag{13}$$

In the next step, inflow segments with $\Delta Q_j^c > 0$ and outflow segments with $\Delta Q_j^c < 0$ can be identified and indexed. For the example in Fig. 4 we index starting from $S_{\text{min}}$: $Q_{\text{out},1}^c = \Delta Q_1^c$, $Q_{\text{in},1}^c = \Delta Q_2^c$, $Q_{\text{out},2}^c = \Delta Q_3^c$, and $Q_{\text{in},2}^c = \Delta Q_4^c$. The representative salinities are calculated for each inflow and outflow similar to (4):

$$s_{\text{in},m} = \frac{Q_{\text{in},m}^s}{Q_{\text{in},m}}, \quad s_{\text{out},m} = \frac{Q_{\text{out},m}^s}{Q_{\text{out},m}}, \tag{14}$$

where $m$ denotes the index with $m = 1, 2, \dots$ For a classical estuary, (13) reads as (7), where the only dividing salinity except $S_{\text{min}}$ or $S_{\text{max}}$ is $S_{\text{div}} = S(\max(Q^c))$.

The mixing relations of MacCready et al. (2018) and Burchard et al. (2018b) require only one value each for the inflow properties and outflow properties, respectively. These can be obtained from a multi-layer transect by applying weighted averages, i.e. for the inflowing bulk values:

$$Q_{\text{in}}^c = \sum_m Q_{\text{in},m}^c, \quad c_{\text{in}} = \frac{\sum_m Q_{\text{in},m}^c}{\sum_m Q_{\text{in},m}} = \frac{\sum_m c_{\text{in},m} Q_{\text{in},m}}{\sum_m Q_{\text{in},m}}, \tag{15}$$





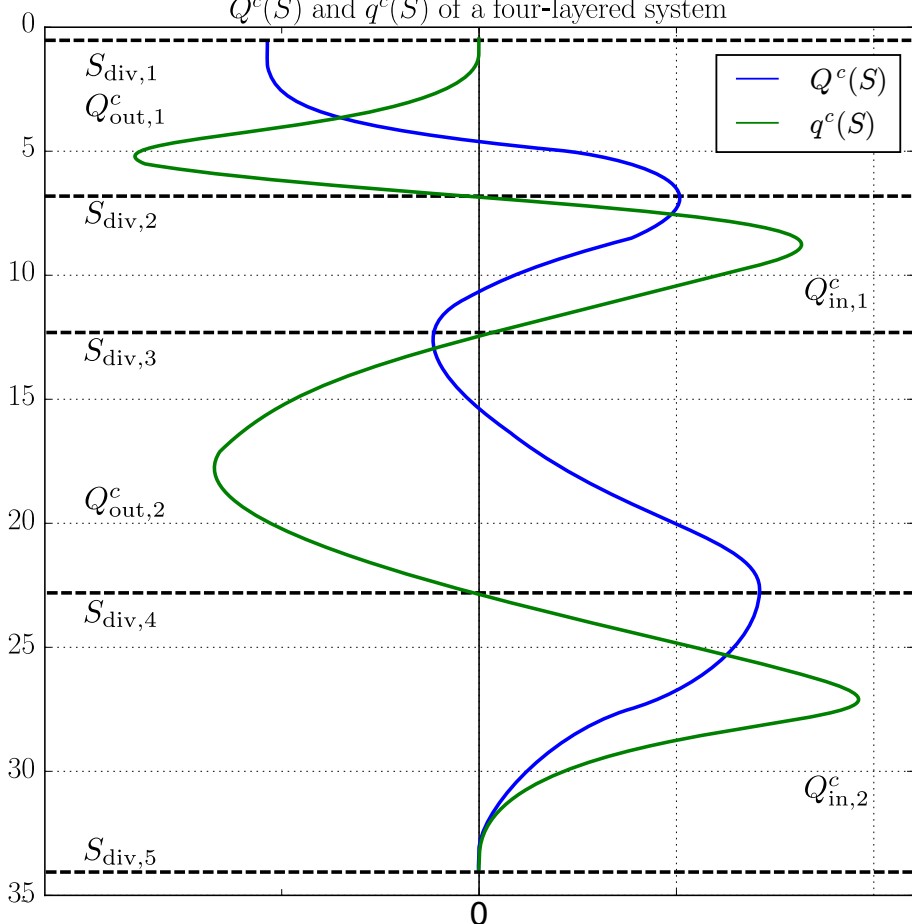

**Figure 4.** Sketch of a hypothetical Total Exchange Flow of a four-layered system with alternating inflows and outflows, $Q^c_{in,m}$ and $Q^c_{out,m}$.
The respective inflows and outflows are divided by the zero crossings of $q^c(S)$ (green), so called dividing salinities, $S_{div,j}$ (dashed, black)
which correspond to the minima and maxima of $Q^c(S)$ (blue).

and accordingly for $Q^c_{out}$ and $c_{out}$.

### 3.2 Discrete formulation

The output from a numerical model along a transect across an estuary is assumed to consist of $I$ time steps with $1 \leq i \leq I$
and $1 \leq k \leq K$ which are spatial increments per each time step. The output should include collocated model data $s^i_k$ (salin-





ity), $c_k^i$ (tracer), and $u_k^i$ (incoming normal velocity) which are available on cross-sectional area increments $A_k^i$. The salinity interval $[S_{1/2}, S_{N+1/2}]$, with $S_{1/2} < S_{\min}$ and $S_{\max} < S_{N+1/2}$, where $S_{\min} = \min(s)$ and $S_{\max} = \max(s)$, is divided into $N$ equidistant intervals of length $\delta S = (S_{N+1/2} - S_{1/2})/N$, compare Fig. 5. The discrete profiles of $q^c$ should be obtained directly without numerically calculating the volume flux profile $Q^c$ before to avoid truncation errors due to numerical derivatives and to save computational time:

$$q_n^c = \frac{1}{I \delta S} \sum_i \sum_{\substack{k \\ (\text{for } n = n_k^i)}} u_k^i c_k^i A_k^i, \quad \text{with} \quad n_k^i = \left\lfloor \left( \frac{s_k^i - S_{1/2}}{\delta S} \right) \right\rfloor, \tag{16}$$

where $\lfloor \cdot \rfloor$ is the integer truncation function. With this, the tracer flux increments are directly added to the respective salinity class, see the dots in the sketch of Fig. 5. Computation of $Q^c(S)$ can be easily carried out by summation of $q_n^c$:

$$Q_{n-1/2}^c = \delta S \sum_n^N q_n^c. \tag{17}$$

Using the extended dividing salinity method defined in (13), the calculation for the transports reads as

$$\Delta Q_j^c = Q_{n=n_{\mathrm{div},j+1}}^c - Q_{n=n_{\mathrm{div},j}}^c, \tag{18}$$

where $n_{\mathrm{div},j}$ and $n_{\mathrm{div},j+1}$ describe the indexes, where two consecutive extrema of $Q^c$ are located. The dividing salinity indices are calculated with an algorithm which searches $Q$ for local extrema by comparing every entry $Q_{n+1/2}$ to its nearest neighbours $Q_{n-1/2}$ and $Q_{n+3/2}$. If $Q_{n+1/2}$ is greater (smaller) than its two neighbours, $n+1/2$ is stored as $n_{\mathrm{div},j}$ and denoted maximum (minimum). Afterwards, transports are computed according to (18) and only dividing salinities with transports greater than a threshold transport $Q_{\mathrm{thresh}}$ are considered. Please see Appendix B for a detailed description.

## 4 Application to exchange flow in the Baltic Sea

The Baltic Sea, shown in Fig. 6, can be considered as a large estuary with a long-term averaged river run-off of around 16000 $\mathrm{m^3 s^{-1}}$ and about balanced precipitation and evaporation (Matthäus and Schinke, 1999). In the estuarine classification diagram by Geyer and MacCready (2014), the Baltic Sea has been classified as a fjord-type and a strongly stratified estuary, due to its relatively low run-off and relatively low mixing. The topography of the Baltic Sea consists of several basins of which the Gotland Basin in the central Baltic Sea, denoted as GB in Fig. 6, is the largest with a water depth of about 240m. The shallow and narrow Danish Straits in the south-west provide the only connection to the saline North Sea.

Episodic inflow events of water consisting of a mixture of saline North Sea water and recirculated brackish Baltic Sea water (Meier et al., 2006) transport large amounts of salt and oxygen into the Baltic Sea. These inflows may either occur as Major Baltic Inflows (MBIs, i.e. as well-mixed, barotropic inflows) during winter months (Matthäus and Schinke, 1999; Mohrholz et al., 2015), or as baroclinic summer inflows (Feistel et al., 2004, 2006). These large inflow events propagate as dense bottom currents from basin to basin, where they are subject to entrainment of overlaying less saline water. The volume of the inflows increases and their salinity decreases on the way into the Central Baltic Sea, where they ventilate the typically



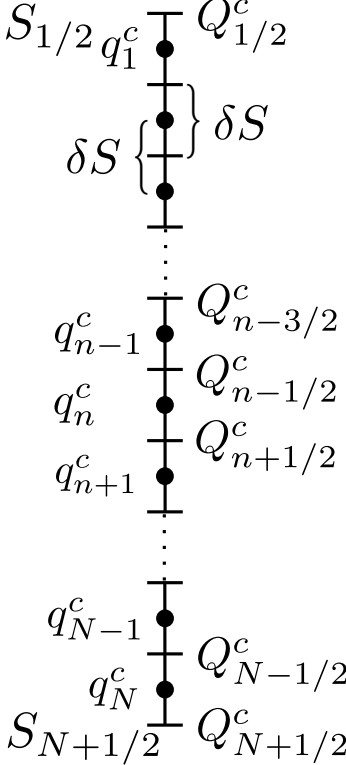

**Figure 5.** Sketch of how $Q^c$ and $q^c$ are located in a discrete salinity space. The salinity interval $[S_{1/2}, S_{N+1/2}]$ is divided into $N$ equidistant salinity classes of length $\delta S$. The entries of $Q^c$, $Q_n^c$, are located on the lines, and the entries of $q^c$, $q_n^c$, are located on the dots.

anoxic bottom layers (Reissmann et al., 2009). More frequent but weaker and less saline inflow events propagate through the Western Baltic Sea (Sellschopp et al., 2006; Umlauf et al., 2007) and have the potential to ventilate intermediate layers but not the bottom layers in the Central Baltic Sea (Reissmann et al., 2009). The major mixing process to transport saline bottom waters towards the surface of the Central Baltic Sea has been identified as boundary mixing (Holtermann et al., 2012, 2014).

5  However, recently double diffusion in the stratified interior has been discussed as another possibly efficient mixing process in the Baltic Sea (Umlauf et al., 2018). Finally, various surface mixed layer processes mix the salt into the surface layer of the Baltic Sea, such that a horizontal surface salinity gradient is estblished, with salinities varying from 25 g/kg in the Kattegat (K) to 5 g/kg in the Bothnian Bay (BoB). A permanent halocline separates these surface waters from the saline bottom waters. The halocline is located approximately in 70-90 m depth in the Gotland Basin. In addition, a seasonal thermocline develops during

10  summer between 10-30 m (Reissmann et al., 2009). At times, salinity inversions occur in the strongly stratified thermocline, with surface waters being slightly more saline than waters in the thermocline (Burchard et al., 2017).





Above the halocline, driven by wind, inflows and Earth rotation, a cyclonic circulation is generally present in the Central Baltic Sea, with net northward flow in the east of Gotland and southward flow in the west of Gotland (Meier, 2007; Omstedt et al., 2014). This cyclonic circulation is also present in the deeper layers of the Central Baltic Sea, possibly driven by inflows and boundary mixing processes (Hagen and Feistel, 2007; Meier, 2007; Holtermann and Umlauf, 2012). This deep-water mean circulation is overlaid by topographic waves and inertial oscillations (Holtermann et al., 2014).

In the following, the numerical properties of the TEF analysis framework are tested against two transects of the Baltic Sea. The first transect is located across Darss Sill, (D, red transect), in the western Baltic Sea over which part of the exchange with the North Sea is occurringi, see Sect. 4.1. The second transect (green) is located in the Gotland Basin where we apply the extended dividing salinity method to the complicated multi-layer current system, see Sect. 4.2.

## 4.1 Exchange flow over Darss Sill

In their recent review paper, Burchard et al. (2018a) applied the Knudsen relations and the TEF analysis framework to analyse 65 years of high-resolution numerical model output for the Western Baltic Sea using GETM (Burchard and Bolding, 2002; Hofmeister et al., 2010; Klingbeil and Burchard, 2013). Here, we investigate numerical properties of the TEF calculations based on the same numerical model output for the complex inflow years 2002/2003 with several barotropic and baroclinic inflows (Feistel et al., 2006) over the Darss Sill transect shown in Fig. 6.

The horizontal resolution of the model is about $600\,\mathrm{m}$, and the water column is discretised by 42 vertical adaptive layers, the thickness of which vary in time and space (Gräwe et al., 2015). The salinity, velocity and layer thickness data are interpolated to 95 locations equally spaced by $\Delta x = 545$ m along the 52 km long Darss Sill transect which is directed in northwest-southeast direction, such that the number of data points per time step is $K = 42 \cdot 95 = 3990$. The model output time step is $\Delta t = 3$ h, such that $I = 5840$ time steps for two simulation years are stored. These 3-hourly values are consistently averaged using all baroclinic time steps and the Thickness-Weighted Averaging method (Klingbeil et al., 2018b).

Application of the TEF analysis framework for $N$ different salinity classes is shown in Fig. 7, where a classical two-layer exchange flow with inflow at high salinities is seen. The upper panels show $q$ and the respective TEF bulk values, computed with the sign method. $q$ becomes more noisy with increasing $N$ and causes the sign method to converge towards the absolute exchange values. The bulk values change considerably with increasing $N$. The lower panels show $Q$ for the same $N$ and the TEF bulk values computed with the extended dividing salinity method. These bulk values do converge for increasing $N$ towards constant values. For this case $Q_{\mathrm{thresh}}$ was set to $Q_{\mathrm{thresh}} = 100\,\mathrm{m}^3\mathrm{s}^{-1}$.

The values found in here with the dividing salinity method confirm that the found bulk values in Burchard et al. (2018a) are correct and did not experience great errors from using the sign method.

Similar to the dependency of the TEF bulk values on $I$ in Sect. 2, we investigate the dependency on the frequency of model output $\Delta t$ in Fig. 7. For that we averaged the raw 3-hourly output again using Thickness-Weighted Averaging to correspond to coarser time steps $\Delta t$ and smaller $I$ before applying TEF. For the dividing salinity method the relative differences to estimated reference values for different time steps of the model output are calculated. The reference bulk values have been calculated by the dividing salinity method for $N = 2^{16} = 65536$ salinity classes and the 3-hourly output, since the exact values are not



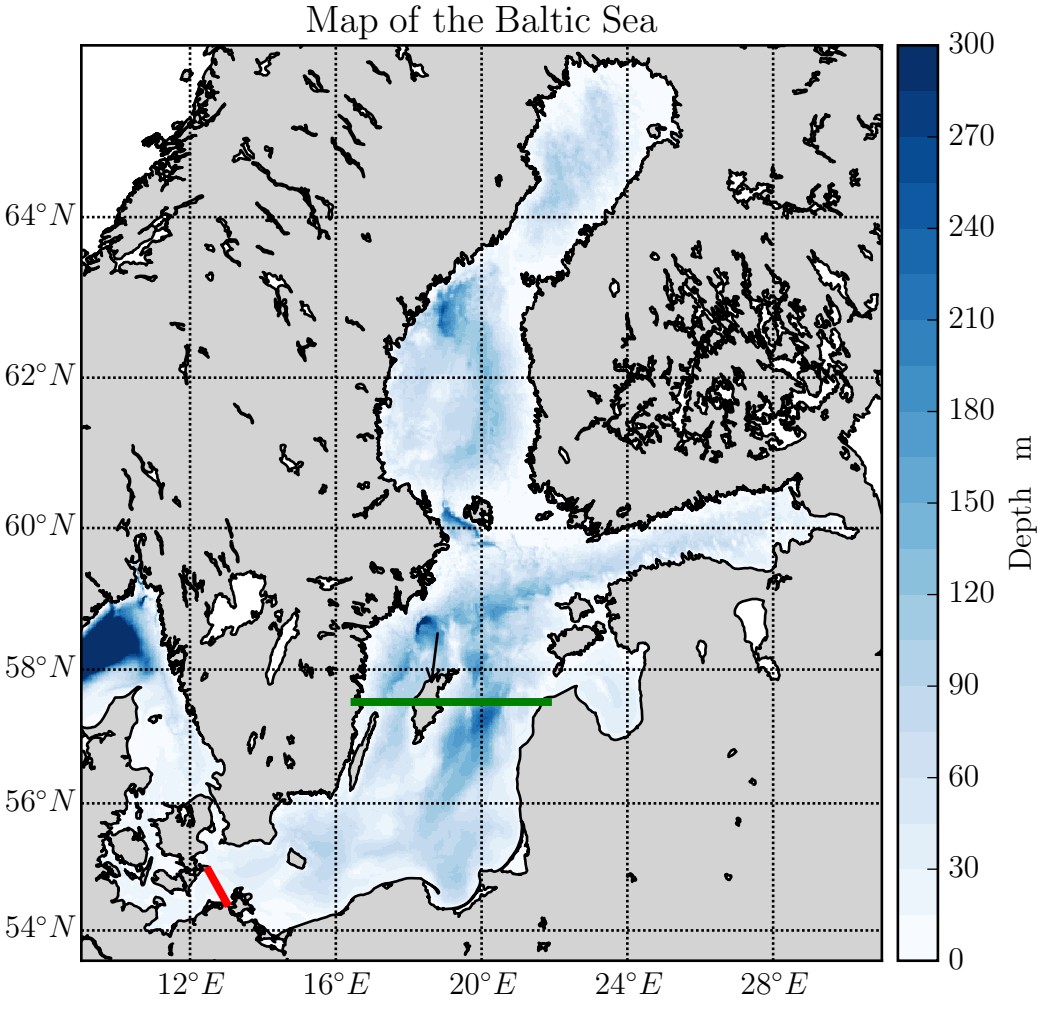

**Figure 6.** Map and bathymetry of the Baltic Sea. K: Kattegat, D: Darss Sill and the Darss Sill transect (red), G: The island Gotland, GB: Gotland Basin and the Gotland transect (green), BoB: Bothnian Bay.





available. The model is forced with 3-hourly data, meaning that external processes of smaller time scales are not included. Therefore, the estimated bulk values can be considered as good estimations. Fig. 8a shows $Q(S, \Delta t)$ with the corresponding dividing salinities. With coarser temporal resolution (larger $\Delta t$), the maximum of $Q$ moves towards greater salinities and smaller transport values, showing a weakened exchange flow. For $\Delta t = 10$d the maximum shifts back to smaller salinities,

indicating that some processes are not resolved anymore. Furthermore, the maximum salinities decrease with reduced temporal resolution which indicates that the inflows of high salinities are not captured. In Fig. 8b the relative deviations of the TEF bulk values are shown for the inflow. With increasing time step $\Delta t$ the deviations increase rapidly as one would expect since processes of smaller time scales are not resolved anymore. For $\Delta t \geq 3$d the deviations fluctuate around a constant value with the exception of $\Delta t = 5$d. The deviations for this time step are smaller than expected. Fig. 8a shows that the shape of $Q(S, 5\text{d})$ is

closer to the shape of the 3-hourly output, leading to more correct bulk values, which we expect to be accidental. The properties of the outflow follow a similar pattern with generally smaller deviations since the outflow does not depend as much on inflows events, not shown here. Fig. 8b also shows that for this simulation 12-hourly model output is enough to resolve the exchange flow properly, i.e. errors of less than 1%.

## 4.2 Cross section through the Gotland Basin

In this section, the capability of the extended dividing salinity method to be applied to exchange flows or transects with more than two layers, is demonstrated. Here, example results are shown for model data of the Gotland Basin in the Baltic Sea. The analysed transect uses the model run from Burchard et al. (2018a) consisting of 156 equally spaced locations with one nautical mile resolution and 50 vertical adaptive layers. Daily averages from two simulation years, 2002 and 2003, are analysed. These two years show a complex inflow activity, with baroclinic inflows during summer 2002 and summer 2003 and an MBI during

winter 2002/2003 (Feistel et al., 2006).

Fig. 9a shows $q$ for $N = 2^8 = 256$ salinity classes to visualise the exchange flow, whereas Fig. 9b shows $Q$ for $N = 2^{16} = 65536$ which is used to compute the bulk values using the extended dividing salinity method (14) and (18). For this dataset five dividing salinities are found, using $Q_{\text{thresh}} = 0.01 \cdot \max(|Q|) \approx 700 \text{ m}^3\text{s}^{-1}$, separating two inflows $Q_{\text{in},1}$ and $Q_{\text{in},2}$, and two outflows, $Q_{\text{out},1}$ and $Q_{\text{out},2}$. These are listed with their respective salinities $s_{\text{in},1}$, $s_{\text{in},2}$, $s_{\text{out},1}$ and $s_{\text{out},2}$ on the right of Fig. 9 for

$N = 2^{16}$ salinity classes.

The net southward transport of $11300 \text{ m}^3\text{s}^{-1}$ results from the fact that most river input is entering the Baltic Sea north of the transect. $Q_{\text{in},1}$ and $Q_{\text{out},1}$ belong to the cyclonic surface circulation of the Gotland Basin described above. With the main river input in the north the outflow $Q_{\text{out},1}$ is less saline than the inflow $Q_{\text{in},1}$ which experiences more entrainment of saline bottom waters during the recirculation. $Q_{\text{in},2}$ describes the net northward transport of the deep circulation which is fed

with high salinities of the inflow events. $Q_{\text{out},2}$ is the corresponding deep net southward transport of less saline water which is homogeneous over a salinity range from $\sim 8$ to $\sim 10$ g/kg, see Fig. 9a. Further and more detailed TEF analyses of the dynamics in the Gotland Basin should be carried out in the future but will be not part of this study, as the focus lies on the method and not the physics. Nevertheless, the extended dividing salinity method proves to be suitable to find robust bulk values for multi-layered exchange flows.




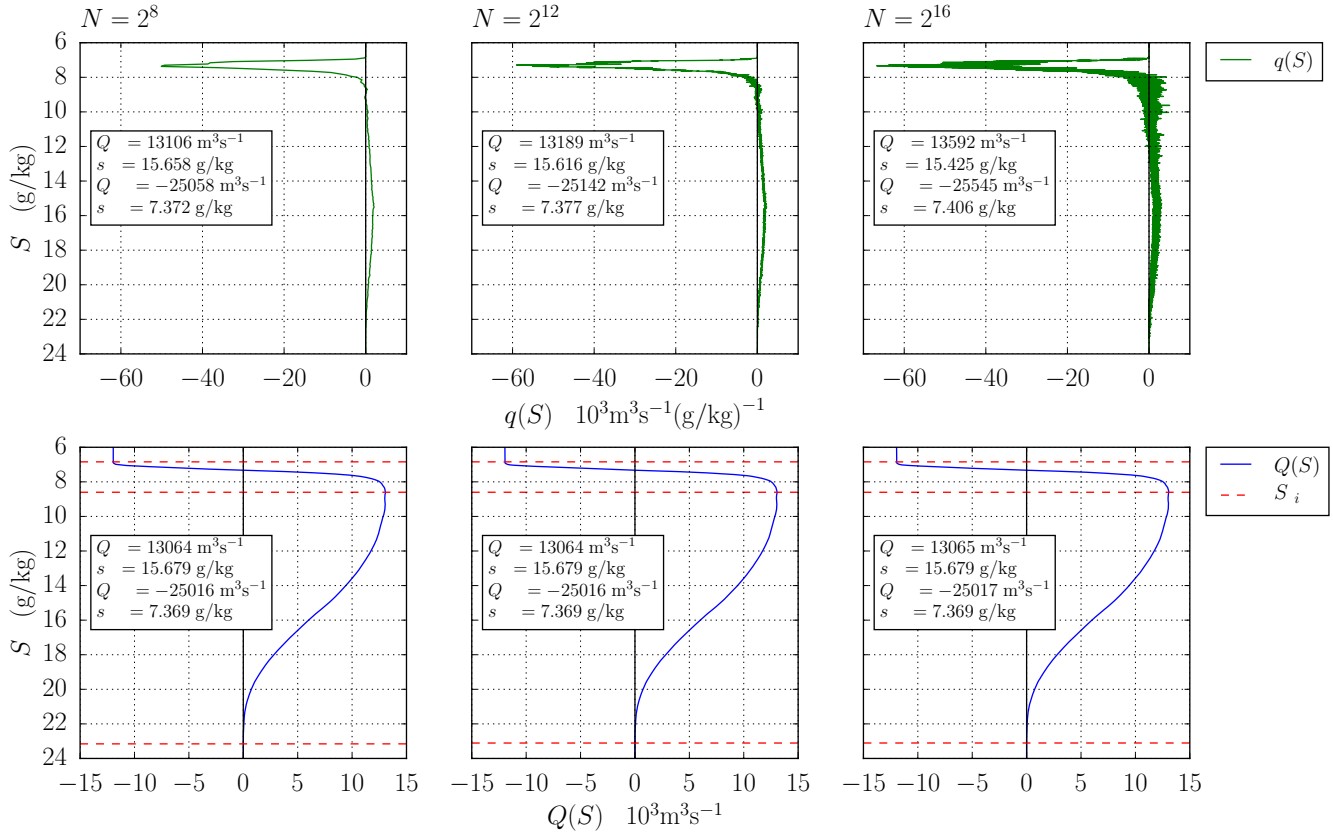

**Figure 7.** Profiles of $q$ (upper panels) and $Q$ (lower panels) for the Darss Sill transect in 2002/2003 in dependency of the number of salinity classes $N$, a) and d) $N = 2^8 = 256$, b) and e) $N = 2^{12} = 4096$, c) and f) $N = 2^{16} = 65536$. The respective TEF bulk values are calculated with the sign method (3) in the upper panels and the extended dividing salinity method (14), (18) in the lower panels.

## 5  Discussion and Conclusions

This study investigated the numerical issues of the Total Exchange Flow (TEF) analysis framework, proposed by MacCready (2011). Two existing calculation methods for the computation of the bulk values of an exchange flow, the *sign method* ((3) MacCready (2011)) and the *dividing salinity method* ((7) MacCready et al. (2018)) were compared in their respective con-
5   vergence behaviours for an analytical test case. We could show that only the dividing salinity method converges towards the analytical bulk values. The sign method relies on a smooth $q$ profile, but $q$ tends to become more noisy with increasing number of salinity classes (for constant temporal resolution), which leads to wrong convergence to absolute values (6). The dividing salinity method on the other hand relies on a smooth $Q$. Although $q$ is very noisy for a high number of salinity classes, $Q$ allows a convergent and robust calculation of TEF bulk values. An extended formulation of the dividing salinity method is presented
10   which includes exchange flows of more than two layers as well as inverse exchange flows. We showed the application to two


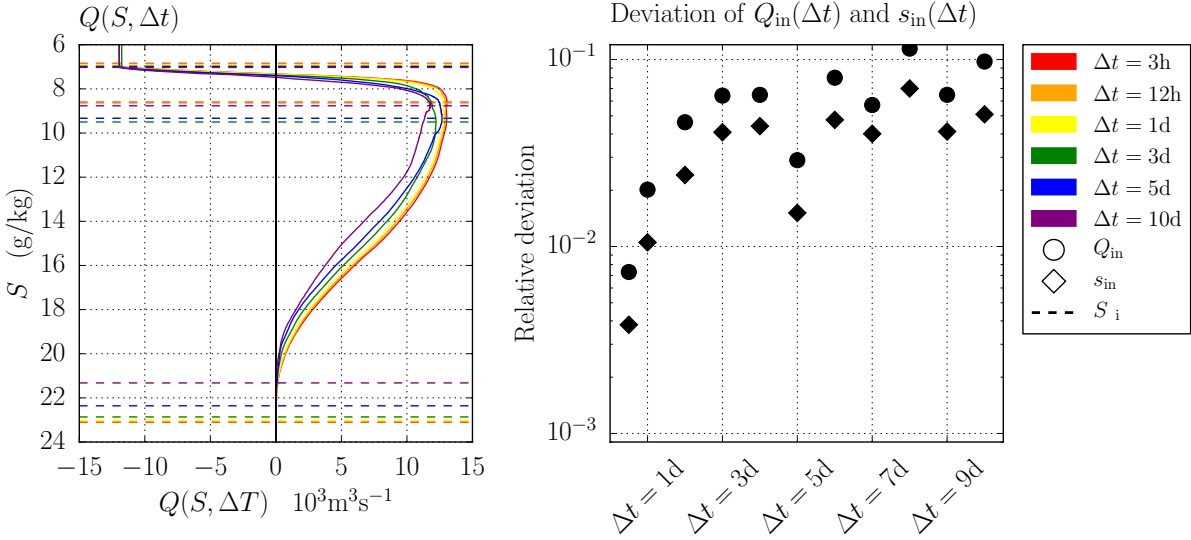

**Figure 8.** Comparison of $Q(S)$ ($N = 2^{16} = 65536$) for different $\Delta t$ in a) and the relative deviations of $Q_{\mathrm{in}}$ and $s_{\mathrm{in}}$ in dependency $\Delta t$. The bulk values we computed from $Q(\Delta t)$ using the extended dividing salinity method (13),(18). The dashed lines in a) show the dividing salinities used to compute the bulk values in b). With different temporal resolutions the shape of $Q(S)$ changes considerably and the resulting bulk values deviate significantly from the estimated ones.

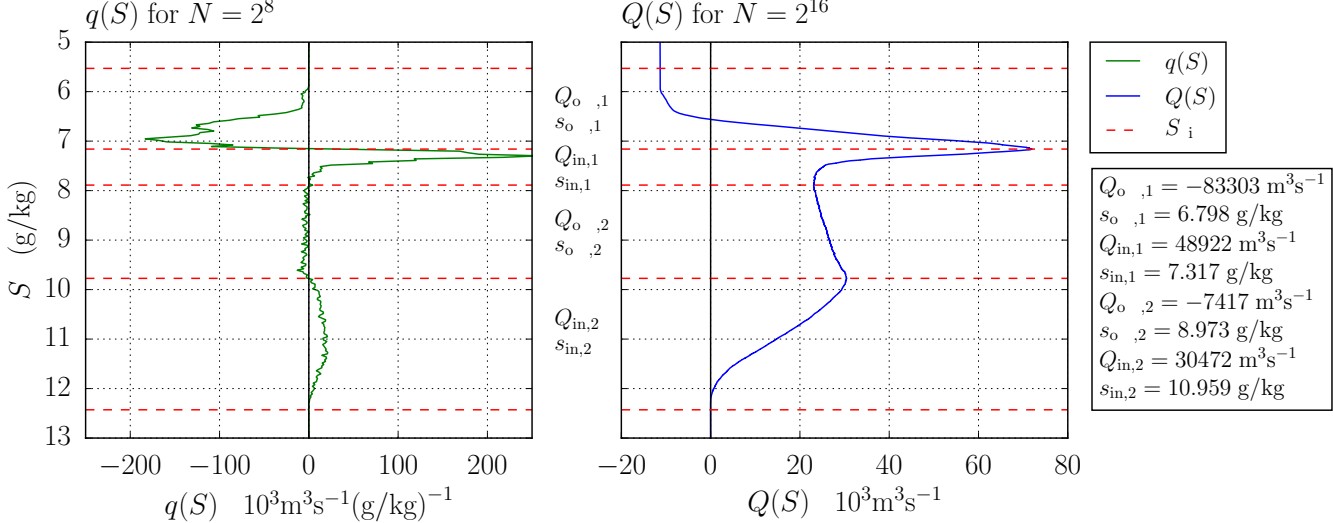

**Figure 9.** Profiles of $q$ for $N = 2^8 = 256$, a), and $Q$ for $N = 2^{16} = 65536$, b), for the Gotland transect in 2002/2003. Five dividing salinities separate two inflows and two outflows. The corresponding TEF bulk values are listed on the right.




transects of the Baltic Sea. The main challenge of the extended dividing salinity method is finding the dividing salinities. For that, one needs a robust algorithm which finds the extrema of $Q$. The algorithm we came up with is described in detail in Appendix B. Moreover, we investigated the dependency of the calculated bulk values on the frequency of model output. The results confirm that the output of the model for a transect which should be analysed by the application of TEF is strongly

5    dependent on the physical mechanism controlling the exchange flow.

Based on our results we propose a best-practise procedure for calculating TEF from a numerical model:

1. At the level of setting up a numerical model, the spatial (horizontal and vertical) resolution should be chosen as high as possible to reproduce return flows due to lateral eddies and smaller overturns.

2. Once a transect for the TEF analysis has been identified, the frequency for storing the output along that transect should be high enough to resolve the driving mechanisms of the exchange flow. Ideally, either results for all baroclinic time steps would be stored or the numerical model should do the binning into salinity classes of a chosen transect itself and save profiles of $q^c$ for desired tracers $c$.

3. Required output fields are the velocity component normal to the transect, the salinity and the grid box area along the transect. We suggest that these variables are stored as Thickness-Weighted Averaged values (Klingbeil et al., 2018a) between two saving time steps to ensure the conservation of volume and salinity.

4. The results should be analysed for a large range of salinity classes $N$ with the dividing salinity method ((13) and (14)) to check the convergence of the TEF bulk values. In this study $N \approx 1000$ salinity classes, $\sim \delta S = 0.02$ g/kg, were sufficient enough for all three investigated examples with errors or deviations smaller than $0.1\%$.

5. Visualisation of the exchange flow should still be done with a smooth $q$ since it shows the inflows and outflows more clearly. We suggest to choose $N \approx 250$ for estuaries with a wide range of salinities or a step size in salinity space of $\sim 0.05$ g/kg, i.e. 20 steps per 1 g/kg, for estuaries with smaller salinity ranges.

## Appendix A: Analytical solution for $Q(S)$ and $Q^s(S)$

For the oscillating exchange flow given in (9), the analytical solution is given here for the volume flux profile $Q(S)$ and the salinity flux profile $Q^s(S)$. According to (1), these profiles are calculated as

$$Q(S) = \left\langle \int_{A(S)} u\, dA \right\rangle = \frac{A}{T} \int_{t^{(1)}(S)}^{t^{(2)}(S)} u(t)\, dt$$

$$= \frac{A}{\omega T} \left[ u_r \omega t + u_a \sin(\omega t) \right]_{t^{(1)}(S)}^{t^{(2)}(S)}$$

(A1)




and

$$Q^s(S) = \left\langle \int_{A(S)} u\,s\,\mathrm{d}A \right\rangle$$

$$= \frac{A}{T} \int_{t^{(1)}(S)}^{t^{(2)}(S)} u(t)\,s(t)\,\mathrm{d}t \tag{A2}$$

$$= \frac{A}{\omega T} \left[ u_r s_r \omega t + u_a s_r \sin(\omega t) + u_r s_a \sin(\omega t + \phi) \right.$$

$$\left. + \frac{u_a s_a \cos(\phi)}{2}\left(\omega t + \sin(\omega t)\cos(\omega t)\right) - \frac{u_a s_a \sin(\phi)}{2}\sin^2(\omega t) \right]_{t=t^{(1)}(S)}^{t=t^{(2)}(S)},$$

with

$$t^{(1)}(S) = -\frac{1}{\omega}\left(\arccos\left(\frac{S-s_r}{s_a}\right) + \phi\right), \quad t^{(2)}(S) = \frac{1}{\omega}\left(\arccos\left(\frac{S-s_r}{s_a}\right) - \phi\right), \tag{A3}$$

5   which ensures that $s(t) \geq S$ for $t^{(1)}(S) \leq t \leq t^{(2)}(S)$ and $s(t) < S$ for $t^{(2)}(S) < t < t^{(1)}(S) + T$. $q(S)$ is calculated according to (2):

$$q(S) = \frac{A}{\omega T \sqrt{s_a^2 - (S-s_r)^2}}\left[ u\left(t^{(1)}\right) + u\left(t^{(2)}\right) \right]$$

$$= \frac{2A}{\omega T \sqrt{s_a^2 - (S-s_r)^2}}\left[ u_r + u_a \frac{S-s_r}{s_a}\cos(\phi) \right] \tag{A4}$$

The dividing salinity can be calculated by finding the root of $q(S)$. Solving (A4) with $q(S_{\mathrm{div}}) = 0$ for $S_{\mathrm{div}}$:

$$S_{\mathrm{div}} = \frac{-s_a u_r}{u_a \cos(\phi)} + s_r. \tag{A5}$$

10   The TEF bulk values can be calculated according to (7) and (4).

**Appendix B: Algorithm description**

The algorithm finding the extrema of $Q$ works as follows. First, every entry of $Q_{n+1/2}$ of $Q$ is compared with its nearest neighbours $Q_{n-1/2}$ and $Q_{n+3/2}$. If $Q_{n+1/2}$ is either the maximum (minimum) in this interval, the index $n+1/2$ is stored and denoted by max (min), respectively. Afterwards, consecutive maxima or minima are deleted, leaving only the greatest

15   maxima or the smallest minima. Now, minima and maxima should be alternating. At this stage there are probably physically





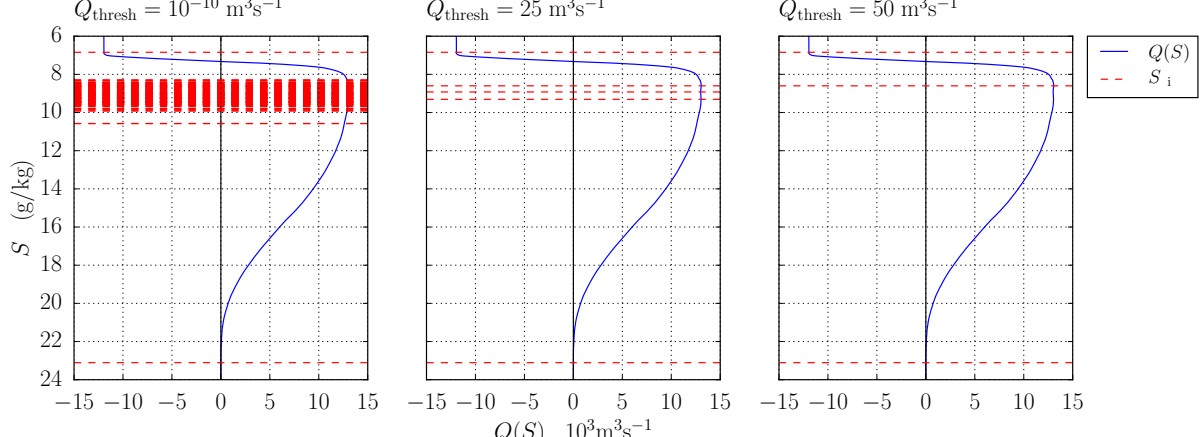

**Figure B1.** Comparison of the algorithm for a) $Q_{\text{thresh}} = 10^{-10}$ m$^3$s$^{-1}$, b) $Q_{\text{thresh}} = 25$ m$^3$s$^{-1}$ and c) $Q_{\text{thresh}} = 25$ m$^3$s$^{-1}$ for the Darss Sill data with $N = 4096$.

insignificant extrema found. Therefore, transports are calculated according to (18), their absolute values $|\Delta Q_j|$ are compared to a given threshold value $Q_{\text{thresh}}$, which we recommend to set to a value around $0.01 \cdot \max(|Q|)$ m$^3$s$^{-1}$. If the transport $|\Delta Q_j|$ is smaller than $Q_{\text{thresh}}$, $Q(S_{\text{div},j})$ and $Q(S_{\text{div},j+2})$ are compared and only the greater (smaller) of the two is kept to ensure that the greater maxima (smaller minima) remains. The two dividing salinities which belong to the smaller (greater) transport are

then not considered anymore. If the first or last extremum is involved in this procedure, only the extrema which is not the first or last extrema is deleted. If this needs to be done, then the first or last extrema changes its property from either minimum to maximum or the other way round to ensure alternating minima and maxima. The last step is to adjust the first and last extrema to the index where $Q_{n+1/2}$ starts to differ from $Q_{1/2}$ (low salinities) or where $Q_{n+1/2}$ differs from 0 (high salinities). This step is not necessary for calculating the correct TEF bulk values since only the dividing part is important and not the exact value of

the dividing salinity. Nevertheless, this procedure ensures that $S_{\text{div},1}$ is the salinity class next to $\min(s)$ and $S_{\text{div},J+1}$ is next to $\max(s)$, with $J$ being the number of layers.

Figure B1 shows the sensitivity of the number of dividing salinities on $Q_{\text{thresh}}$ for the data from Section 4.1 for $N = 4096$ salinity classes. In Fig. B1a for $Q_{\text{thresh}} = 10^{-10}$ m$^3$s$^{-1}$ (to filter out numerical noise of double precision data) 135 dividing salinities, most between 8 and 10 g/kg are found. Most of them are noise carried on from the $q$ profile to $Q$ and have no

physical meaning. However, two major transports are found with $-24885$ and $12603$ m$^3$s$^{-1}$. For $Q_{\text{thresh}} = 25$ m$^3$s$^{-1}$ noise related transports are filtered out, leaving two small transports of 63, and $-44$ m$^3$s$^{-1}$. The two main transports change to $-25016$ and $13045$ m$^3$s$^{-1}$. Increasing to $Q_{\text{thresh}} = 50$ m$^3$s$^{-1}$, the $-44$ m$^3$s$^{-1}$ is not accounted and according to the algorithm the two involved dividing salinities are deleted. This deletes the 63 m$^3$s$^{-1}$ transport as well. As a result the net transport of 19 m$^3$s$^{-1}$ transport is now accounted to the major inflow, which increased from 13045 to 13064 m$^3$s$^{-1}$ if compared to Fig. B1b.

These are the exact same results as Fig. 7b, where $Q_{\text{thresh}} = 100$ m$^3$s$^{-1}$ was used.



*Competing interests.* The authors declare that they have no conflict of interest.

*Acknowledgements.* This paper is a contribution to BMBF-GROCE FKZ 03F0778. H.B. and M.L. were supported by Research Training
Group Baltic TRANSCOAST GRK 2000 funded by the German Research Foundation. K.K. was supported by the Collaborative Research
Centre TRR 181 on Energy Transfer in Atmosphere and Ocean funded by the German Research Foundation and P.M. was supported by US
5   National Science Foundation grant OCE-1736242.



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
