# Peer review of "Numerical issues of the Total Exchange Flow (TEF) analysis framework for quantifying estuarine circulation"

_Ocean Science, 2018_

## Referee Comment (RC1) · Anonymous Referee #1 · 4 Mar 2019

This is a very useful manuscript that provides a detailed comparison of two methods to calculate the total exchange flow from numerical simulations. The paper ends with a best practice recipe to do these calculations. I would recommend that this paper be published essentially in its current form. I do have a few minor questions which perhaps they can address in the final version.

1) While the authors suggest that the " dividing salinity method" is preferred to the "sign method" but the former requires a algoritm to find extrema of Q. While they provide a detailed description of the algorithm they :"came up with", that particular working ("that we came up with") made me wonder if they feel there are shortcomings in this

method. If so–please elaborate. If not–perhaps they could change the wording to something like. ..” We provide a detailed description of an algorithm to obtain extrema of Q which is required to determing the dividing salinity values”...

2) Section 4.'1 line 25. “The bulk values change considerably”–I assume they mean s, Q. While the do show more variability than the dividing salinity method–they only vary by a few percent... so referring to it as considerable change seems a bit severe. Also shouldn't the bulk quantities be noted as Qin,Sin, Qout and Sout?

3)The map does not include the indicating the locations of places mentioned in the text ( Gotland Island, Gotland Basin, Bothnian Bay)

---

## Referee Comment (RC2) · Anonymous Referee #2 · 8 Mar 2019

The MS is devoted to practical implementation of Total Exchange Flow (TEF) analysis framework, which extends the applicability of historical Knudsen relations to real unsteady space-varying estuarine flows. The MS introduces an interesting analytical example that is used to analyze convergence of two slightly different TEF methods. It is shown, that "dividing salinity" method provides less noisy results than "sign" method in evaluating the volume flux of a specific salinity class from a large set of data. The "dividing salinity" method has been extended to multiple inflow and outflow ranges, allowing more than one dividing salinity. The method has been tested based on the Baltic Sea modelled data. Recommendations are also given to choose the parameters for proper estimation of TEF. The results of the MS are interesting and of significant

practical importance.

Essential part of the MS is devoted to finding optimum number of salinity bins in order to discretize the problem with acceptable noise level. The parameter search was done by a large number of trials, the essence of which is not easy to understand. MacCready (2011) noted in his founding paper of the TEF method, that discretization is analogous to finding histogram of transport versus salinity. Selection criteria of the number of bins are well known for histograms. Drawing parallels between traditional histogram bins and TEF salinity bins would hopefully improve readability of the paper for wider audience.

The central finding in the MS is that one of the TEF calculation methods, the "sign" method does not converge to the analytically determined exchange flow when the number of data points and the number of salinity bins both increase. The other method, the "dividing salinity" method reveals nice convergence and therefore is the preferred TEF calculation method. This result is mentioned in several places, but proof is presented only very shortly in Section 2 P4 L25 to P5 L7 based on Fig. 3. This figure contains many curves and is not easy to read. Alternative or complementary figures reflecting the conclusion should be welcome. Another issue with convergence presentation appears in the introduction on P3 L5-9 and L14-18 with infinite number of salinity bins. It remains unclear if eq (5) is mathematically derived or guessed from numerical experiments. The detailed parts could be moved to Section 2 in order to support analysis of the convergence of numerical results to analytical findings.

Proper selection of sampling interval (in present case, interval of model output) is very important for determination of Eulerian volume transports and salinity-space exchange flows, using reasonable computational resources. This aspect is shortly presented in Section 4.1 with an example of flows over Darss Sill in the Baltic Sea, mainly describing the curves in Fig. 8. The 3-hourly output values were averaged using a special method that description is not available at the moment. At the end of the section it is written "12-hourly model output is enough to resolve the exchange flow properly". The conclusions

say on the same matter "Ideally, either results for all baroclinic time steps would be stored or the numerical model should do the binning into salinity classes of a chosen transect itself and save profiles of qc for desired tracers c". It is not clear, how the conclusions match the findings in the previous chapters.

Some specific questions and comments are presented below.

P1 L11-12: The sentence "Since inflow and outflow occurring at the same salinity compensate, TEF characterises the net exchange flow with the ambient ocean" should be reformulated. The first part of the sentence could be understood that the net exchange flow is missing since inflow and outflow compensate. Perhaps it should be "Since oscillatory inflow and outflow components occurring at the same salinity compensate...".

P3 L2: "TEF profiles computed from numerical model output can be noisy", it should be useful to present some physical reasons of noisy results. Perhaps, within well-mixed space-time domains conversion from depth to salinity coordinates has errors. Other reasons may also exist.

P3 L19: "Obviously, this dividing salinity method only works for classical exchange flows". Please consider if this sentence is needed at this point of introduction.

P4 L13: "To visualise why only the dividing salinity method is converging towards the real bulk values" is not clear, "why only" cannot be found. It is not explicitly written by which method Fig. 2a-c have been produced.

P4 L24: "The differences of the noises are because of ...", it is not clear the noises of what quantities are considered. It is well known that integrating the discrete curve makes the result smoother, but taking derivatives amplifies the small-scale variations (noise).

P11 L23-25 : "...TEF bulk values, computed with the sign method. q becomes more noisy with increasing N and causes the sign method to converge towards the absolute exchange values.". It is not clear what the absolute exchange values are and how the
convergence is proved.

I recommend publishing of the MS. I have given some recommendations of minor rewriting in order to improve the clarity and readability.
* * *

---

## Author Comment (AC1) · 23 Apr 2019

Thanks to the anonymous referees for their useful comments. We will go over each point in the following. The equations, pages and lines we refer to in our responses are from the revised manuscript. Supplement material: LaTeX difference of the original and the revised manuscript.

[Figure]

**1   Referee 1**

1) While the authors suggest that the "dividing salinity method" is preferred to the "sign method" but the former requires a algorithm to find extrema of Q. While they provide a detailed description of the algorithm they: "came up with", that particular working ("that we came up with") made me wonder if they feel there are shortcomings in this method. If so, please elaborate. If not, perhaps they could change the wording to something like..." We provide a detailed description of an algorithm to obtain extrema of Q which is required to determining the dividing salinity values" ...

In principle, any algorithm dedicated to finding extrema should be suitable for the first step of finding the extrema of $Q$. It is then important that the final extrema are alternating minima and maxima and to reduce the extrema to the relevant ones.

We don't see any shortcomings of using the provided algorithm as it should find any minimum and maximum by using the 3 point window. The key to lower the number of relevant extrema is the threshold transport which filters out the physically unimportant ones.

We agree to rephrase to: "We provide a detailed description of a robust algorithm to obtain extrema of $Q$ which is required to determine the dividing salinities in Appendix B." (P16 L4f)

2) Section 4.1 line 25. "The bulk values change considerably", I assume they mean s, Q. While they do show more variability than the dividing salinity method, they only vary by a few percent. . . so referring to it as considerable change seems a bit severe. Also, shouldn't the bulk quantities be noted as $Q_{in}$, $s_{in}$, $Q_{out}$ and $s_{out}$?

We just recognized that in the pdf provided by Ocean Science all labels from the pdf figures are not displayed correctly. In the original upload they are denoted by $Q_{in}$ etc..

We rephrased to: "The bulk values still change with increasing $N$." and deleted "and causes the sign method to converge towards the absolute exchange values.", because we don't provide the absolute values.

3) The map does not include the indicating the locations of places mentioned in the text (Gotland Island, Gotland Basin, Bothnian Bay).

Indications are also lost due to the pdf error.

Please also note the supplement to this comment:
https://www.ocean-sci-discuss.net/os-2018-147/os-2018-147-AC1-supplement.pdf

[Figure]

**Supplement:**

**Numerical issues of the Total Exchange Flow (TEF) analysis framework for quantifying estuarine circulation**

Marvin Lorenz[1], Knut Klingbeil[1], Parker MacCready[2], and Hans Burchard[1]

[1]Leibniz Institute for Baltic Sea Research Warnemünde, Rostock, Germany
[2]University of Washington, Seattle, Washington

**Correspondence:** Marvin Lorenz (marvin.lorenz@io-warnemuende.de)

**Abstract.** For more than a century, estuarine exchange flow has been quantified by means of the Knudsen relations which connect bulk quantities such as inflow and outflow volume fluxes and salinities. These relations are closely linked to estuarine mixing. The recently developed Total Exchange flow (TEF) which uses salinity coordinates to calculate these bulk quantities allows an exact formulation of the Knudsen relations in realistic cases. There are however numerical issues, since the original method does not converge to the TEF bulk values for an increasing number of salinity classes. In the present study, this problem is investigated and the method of *dividing salinities*, described by MacCready et al. (2018), is mathematically introduced. A challenging yet compact analytical scenario for a well-mixed estuarine exchange flow is investigated for both methods, showing the proper convergence of the dividing salinity method. Furthermore, the dividing salinity method is applied to model results of the Baltic Sea to demonstrate the analysis of realistic exchange flows and exchange flows with more than two layers.

**1 Introduction**

The *Total Exchange Flow* (TEF) analysis framework calculates time-averaged net volume and mass transports between enclosed volumes of the ocean and ambient water masses, sorted by salinity classes. Since oscillatory inflow and outflow components occurring at the same salinity compensate, TEF characterises the net exchange flow with the ambient ocean. Salinity rather than density or temperature is used as a coordinate for calculating estuarine exchange flow, since only the salt budget is entirely controlled by the exchange flow. Therefore, salt is the only conserved quantity. In contrast, temperature and thus density are additionally affected by the freshwater run-off and the surface heat fluxes.

A first bulk approach based on inflow and outflow salinity and volume transport had been developed and applied to the exchange flow of the Baltic Sea by Knudsen (1900). The theoretical framework based on a continuous salinity space had first been developed by Walin (1977), and later been applied to exchange flow in the Baltic Sea (Walin, 1981). A comparable framework had been applied by Döös and Webb (1994) for quantifying meridional overturning circulation in the Southern Ocean. Both the bulk concept by Knudsen (1900) and the continuous concept by Walin (1977) had been consistently combined by MacCready (2011) who also coined the term *Total Exchange Flow*, TEF.

TEF considers a time-averaged transport of a tracer $c$, $Q^c$, through the cross-sectional area  $A(s > S)$, which has a salinity $s$ above a specific value $S$. $Q^c$ is defined as

$$Q^c(S) = \left\langle \int_{\underline{A(S)}\,\underline{A(s>S)}} c\,u\,\mathrm{d}A \right\rangle,\tag{1}$$

where $u$ is the incoming velocity normal to  $A(s > S)$ with the definition that positive $u$ brings water into the estuary and $\langle\rangle$ denotes temporal averaging. The exchange profile of tracer flux per salinity as  a function of the salinity is then obtained by differentiating $Q^c(S)$ with respect to $S$:

$$q^c(S) = -\frac{\partial Q^c(S)}{\partial S},\tag{2}$$

such that $Q^c$ can be also obtained via integration of $q^c$ in salinity space,

$$Q^c(S) = \int_{S'>S} q^c(S')\,\mathrm{d}S' = \int_{S}^{S_{\max}} q^c(S')\,\mathrm{d}S'.\tag{3}$$

Based on these quantities consistent Knudsen bulk values for inflowing and outflowing salinity ($s_{\text{in}}$, $s_{\text{out}}$), volume flux ($Q^1_{\text{in}} = Q_{\text{in}}$, $Q_{\text{out}}$) and salt flux ($Q^s_{\text{in}}$, $Q^s_{\text{out}}$), obeying

$$s_{\text{in}} = \frac{Q^s_{\text{in}}}{Q_{\text{in}}}, \quad s_{\text{out}} = \frac{Q^s_{\text{out}}}{Q_{\text{out}}},\tag{4}$$

can be obtained. MacCready (2011) calculates the inflowing and outflowing bulk fluxes by integrating over positive and negative parts of $q^c$:

$$Q^{c,\text{sign}}_{\text{in}} = \int_{S_{\min}}^{S_{\max}} (q^c)^+ \,\mathrm{d}S, \quad Q^{c,\text{sign}}_{\text{out}} = \int_{S_{\min}}^{S_{\max}} (q^c)^- \,\mathrm{d}S,\tag{5}$$

where for any function $a$, the positive part is calculated as $(a)^+ = \max(a, 0)$ and the negative part is calculated as $(a)^- = \min(a, 0)$. In (5),  $S_{\min}$ and $S_{\max}$ are the minimum and maximum salinities. We will call this method of integrating positive and negative contributions separately to obtain $Q^c_{\text{in}}$ and $Q^c_{\text{out}}$ *sign method* in the following.

$$s_{\text{in}} = \frac{Q^s_{\text{in}}}{Q_{\text{in}}}, \quad s_{\text{out}} = \frac{Q^s_{\text{out}}}{Q_{\text{out}}}.$$

Recently, Klingbeil et al. (2019) showed the relation between TEF and Thickness Weighted Averaging. The concepts by Knudsen (1900), Walin (1977) and MacCready (2011) were focussed on estuarine systems, which are characterised by distinct volume inflow $Q_r$ of water masses of (almost) zero salinity. The exchange flow between the estuary and the ocean is described by the Knudsen bulk values

 The Total Exchange Flow provides one consistent calculation method for these bulk values, which for this case describe the net exchange flow. Since there is no clear definition of the Knudsen bulk values, we will call these  *TEF bulk values* to distinguish between other bulk values which also fulfill the Knudsen relations, e.g. bulk values computed from a Eulerian version of TEF. The Knudsen relations have been reviewed in detail for exchange flow in the Western Baltic Sea by Burchard et al. (2018a). Recently, MacCready et al. (2018) showed how the bulk concept can be used to estimate the volume-integrated average mixing $M$ (defined as the rate of reduction of the net salinity variance due to mixing) in estuaries: $M \approx s_{\mathrm{in}} s_{\mathrm{out}} Q_r$, i.e. the volume-integrated average mixing in an estuary is approximated by the product of inflow and outflow salinity with the estuarine freshwater supply. This mixing estimate by MacCready et al. (2018) approximates the TEF-based exact formulations developed by Burchard et al. (2018b).

Since the TEF analysis framework is continuous in salinity, a discretisation in salinity space is required when analysing  data from numerical model simulations or field observations. In their Appendix A2,  Klingbeil et al. (2019) p the remapping of discrete data into bins. As a result, the output of a numerical model consists of a finite number of transport values associated with the same number of discrete salinities.  Comparable to a histogram, the transport data are binned into salinity classes according to their associated salinities. As discussed by MacCready et al. (2018), the resulting TEF profiles can become noisy, i.e. sign changes in $q^c$, when the number of discrete salinity classes $N$ is chosen too high. For data sets with pairwise disjunct salinities the number of transport values assigned to a single salinity bin decreases with the number of the salinity bins. After exceeding a threshold number of salinity classes, the bins will be sufficiently small to hold at most one transport value. In this case $Q_{in}^{\mathrm{sign}}$ is equal to $Q_{in}^{\mathrm{abs}}$, with

$$\lim_{N\to\infty} Q_{in}^{\mathrm{sign}}(N) = Q_{in}^{\mathrm{abs}} \neq Q_{\mathrm{in}},$$

$$Q_{\mathrm{in}}^{\mathrm{abs}} = \left\langle \int_A u^+ \mathrm{d}A \right\rangle. \tag{6}$$

 In most practical applications the salinity data are neither constant in space nor time and in the limit of an infinite number of salinity classes $Q_{\mathrm{in}}^{\mathrm{sign}}$ will converge to $Q_{\mathrm{in}}^{\mathrm{abs}}$ which is not the desired result for $Q_{\mathrm{in}}$.

In order to obtain robust bulk values, which are less sensitive to the number of salinity bins, MacCready et al. (2018) suggested an alternative to the sign method

$$Q_{\mathrm{in}}^{\mathrm{abs}} = \left\langle \int_A u^+ \mathrm{d}A \right\rangle.$$

 . Instead of finding an optimal number of bins (a problem well known for histograms (Knuth, 2006)), they suggested to find a *dividing salinity* $S_{\text{div}}$ which separates the inflow and outflow of a classical two-layer estuary with inflow at high and outflow at low salinity classes, i.e. $q^c(S_{\text{div}}) = 0$ and $Q^c(S_{\text{div}}) = \max(Q^c(S))$. The bulk values for inflow and outflow are then obtained by integrating:

$$Q^{c,\text{div}}_{\text{in}} = \int_{S_{\text{div}}}^{S_{\max}} q^c \, \mathrm{d}S, \quad Q^{c,\text{div}}_{\text{out}} = \int_{S_{\min}}^{S_{\text{div}}} q^c \, \mathrm{d}S. \tag{7}$$

It should be noted that analytically and for smooth $q^c$ with only one zero crossing both methods coincide. We will show in Sect. 2 the different convergence behaviours and will show that the *dividing salinity method* indeed converges towards robust TEF bulk values, e.g.

$$\lim_{N \to \infty} Q^{\text{div}}_{\text{in}}(N) = Q_{\text{in}},$$

$\lim_{N \to \infty} Q^{\text{div}}_{\text{in}}(N) = Q_{\text{in}},$ where $Q^{\text{div}}_{\text{in}}$ denotes the  inflowing volume flux computed with the dividing salinity method (7) for $c = 1$.

 Using the maximum of $Q$ only works for classical two-layer exchange flows. In Section 3 we will introduce an extended formulation of the dividing salinity method which includes inverse estuaries (outflow at high salinities and inflow at low salinities) as well as exchange flows with more than two exchange layers in salinity space. Furthermore, in Section 3.2 the corresponding discrete description is presented. Afterwards in Section 4, the extended method is applied to numerical output from a model of the Baltic Sea, before we conclude in Section 5.

**2 Convergence analysis for an analytical classical exchange flow**

To demonstrate the different convergence behaviours of the sign method and the dividing salinity method, we take the analytical example from Burchard et al. (2018b). It describes a well-mixed tidal flow with oscillating salinity as it occurs e.g. in the Wadden Sea (Purkiani et al., 2015). The velocity and salinity are given by

$$u(t) = u_r + u_a \cos(\omega t); \quad s(t) = s_r + s_a \cos(\omega t + \phi), \tag{8}$$

with the residual velocity $u_r < 0$, the residual salinity $s_r$, the velocity and salinity amplitudes $u_a > 0$ and $s_a > 0$ with $s_r - s_a \geq 0$, the tidal frequency $\omega = 2\pi/T$ with the tidal period $T$, and the tidal phase $\phi$. The tidally averaged salinity transport is given by

$$\frac{1}{T} \int_0^T u s \, \mathrm{d}t = u_r s_r + \frac{u_a s_a}{2} \cos(\phi). \tag{9}$$

Zero residual salt transport therefore requires

$$\cos(\phi) = -2 \frac{u_r s_r}{u_a s_a} \quad \text{with} \quad u_a s_a \geq 2|u_r|s_r. \tag{10}$$

Fig. 1 shows an example for $u(t)$, $s(t)$ and $u(t) \cdot s(t)$ with $A = 10000$ m$^2$, $u_r = -0.1$ m s$^{-1}$, $u_a = 1$ m s$^{-1}$, $s_r = 20$ g/kg and $s_a = 10$ g/kg resulting in $\phi = -1.16 = -0.185 \cdot 2\pi$. In this case $Q(S)$, $Q^s(S)$ and $S_{\text{div}}$ can be calculated analytically  by either (5) or (7)  (see Appendix A) and are shown in Fig. 2d. By means of (4), the inflow and outflow volume fluxes and salinities, $Q_{\text{in}}$, $Q_{\text{out}}$, $s_{\text{in}}$, and $s_{\text{out}}$, can then be exactly calculated. The resulting analytical TEF bulk values are $Q_{\text{in}} = 813.240$ m$^3$s$^{-1}$, $Q_{\text{out}} = -1813.240$ m$^3$s$^{-1}$, $s_{\text{in}} = 28.424$ g/kg, and $s_{\text{out}} = 12.748$ 
[revised manuscript text omitted]

---

## Author Comment (AC2) · 23 Apr 2019

Thanks to the anonymous referees for their useful comments. We will go over each point in the following. The equations, pages and lines we refer to in our responses are from the revised manuscript. Supplement material: LaTeX difference of the original and the revised manuscript.

[Figure]

**1 Referee 2**

1) [...] Selection criteria of the number of bins are well known for histograms. Drawing parallels between traditional histogram bins and TEF salinity bins would hopefully improve readability of the paper for wider audience.

Thanks for this very good comment. We incorporated this in the introduction by rephrasing or adding sentences:

- P3L7ff: "Comparable to a histogram, the transport data are binned into salinity classes according to their associated salinities. As discussed by MacCready et al. (2018), the resulting TEF profiles can become noisy, [...]"

- P3L16ff: "In order to obtain robust bulk values, which are less sensitive to the number of salinity bins, MacCready et al. (2018) suggested an alternative to the sign method. Instead of finding an optimal number of bins (a problem well known for histograms (Knuth, 2006)), they suggested to find a *dividing salinity* $S_{div}$ which separates the inflow and outflow of a classical two-layer estuary with inflow at high and outflow at low salinity classes, i.e. $q^c(S_{div}) = 0$ and $Q^c(S_{div}) = \max(Q^c(S))$."

2a) The central finding in the MS is that one of the TEF calculation methods, the "sign" method, does not converge to the analytically determined exchange flow when the number of data points and the number of salinity bins both increase. The other method, the "dividing salinity" method, reveals nice convergence and therefore is the preferred TEF calculation method. This result is mentioned in several places, but proof is presented only very shortly in Section 2 P4 L25 to P5 L7 based on Fig. 3. This figure contains many curves and is not easy to read. Alternative or complementary figures reflecting the conclusion should be welcome.

We took $s_{in}$ out from Fig. 3 and separated the dividing salinity method and the sign method into a and b subplots which makes the figure hopefully more clear and readable, see Fig. 1 below the text.

Caption:

Oscillating exchange flow (see Section 2): relative error of $Q_{in}$ computed with a) the dividing salinity method and b) the sign method in dependency of the number of time steps $I$ (color) and salinity classes $N$. The sign method, (5) and the dividing salinity method, (7) coincide for a small number of salinity classes, but the error of the sign method converges in the limit of large $N$ towards the error of the absolute bulk values (black line, (6)). In contrast, the error of the dividing salinity method converges towards a constant value. The errors of both methods decrease with increasing number of time steps $I$.

2b) Another issue with convergence presentation appears in the introduction on P3 L5-9 and L14-18 with infinite number of salinity bins. It remains unclear if eq (5) is mathematically derived or guessed from numerical experiments. The detailed parts could be moved to Section 2 in order to support analysis of the convergence of numerical results to analytical findings.

We removed the limit against infinity equation and replaced it with a more detailed explanation of why the sign method cannot converge towards the desired bulk values on P3L9-15, but will rather converge towards the absolute values.

"For data sets with pairwise disjunct salinities the number of transport values assigned to a single salinity bin decreases with the number of the salinity bins. After exceeding a threshold number of salinity classes, the bins will be sufficiently small to hold at most one transport value. In this case $Q_{in}^{sign}$ is equal to $Q_{in}^{abs}$, with

$$Q_{in}^{abs} = \left\langle \int_A u^+ \mathrm{d}A \right\rangle. \tag{1}$$

In most practical applications the salinity data are neither constant in space nor time and in the limit of an infinite number of salinity classes $Q_{in}^{sign}$ will converge to $Q_{in}^{abs}$ which is not the desired result for $Q_{in}$."

3) [...] The 3-hourly output values were averaged using a special method that description is not available at the moment. At the end of the section it is written "12- hourly model output is enough to resolve the exchange flow properly". The conclusions say on the same matter "Ideally, either results for all baroclinic time steps would be stored or the numerical model should do the binning into salinity classes of a chosen transect itself and save profiles of qc for desired tracers c". It is not clear, how the conclusions match the findings in the previous chapters.

The "special method" is thickness-weighted averaging (TWA) and is available as accepted JPO manuscript (EOR; doi: 10.1175/JPO-D-18-0083.1).

The comparison of the different methods is valid for any number of time steps. But we wanted to investigate also the influence of the temporal resolution. The results in Section 2 show that the bulk values only match the analytical values, if the temporal resolution is sufficiently high enough, $\sim I = 1000$. We added on P5L2-L5:

"The convergence analysis for different numbers of time steps is done to gain experience in the impact of temporal resolution of the oscillating flow on the final bulk values. With the time step here being the equivalent to the output interval of a hydrodynamic model which provides data for TEF, the findings can directly be transferred to the analysis of model data."

Similarly to the varying $I$ for the analytical scenario, we decreased the number of time points for the model data of Darss Sill by thickness-weighted-averaging to emulate less temporal resolution.

By the definition of $Q$ in (1) the temporal average is outside of the integral which en-

Interactive
comment

sures that all physical, small or large temporal or spatial scales of currents are included. In reality, the model output is saved i.e. 3-hourly which is in itself a conservative mean value (for our model), resulting that the $c$ and $u$ in (1) are mean values. Therefore, $Q$ only approximates the "real" $Q$ of all time steps. If one would save $q$ and $q^s$ online during the model run, there was no need to discuss the frequency of model output and all processes of small timescales were included. The offline analysis adds therefore additional errors. To make this point clearer, we rephrased P16L7-P16L15 in the best-practice section:

"Once a transect for the TEF analysis has been identified, the frequency for storing the output along that transect has to be chosen. For analytical correctness, the binning of data into salinity classes should be done online within the hydrodynamic model at every model time step. Time-averaged model output of these binned data can directly be used for the TEF-analysis. If the model only provides output within the model layers, the binning and averaging must be done offline during postprocessing. This would induce different kind of errors: (i) instantaneous data snapshots which skip intermediate model time steps do not conserve fluxes and do not consider intermediate salinity variations; (ii) model data obtained by thickness-weighted averaging over model time steps conserve fluxes, but merge data of different salinities. Both types of errors can be reduced with a sufficiently high output frequency, such that the output data still resolve the dynamics of the flow."

4) P1 L11-12: The sentence "Since inflow and outflow occurring at the same salinity compensate, TEF characterises the net exchange flow with the ambient ocean" should be reformulated. The first part of the sentence could be understood that the net exchange flow is missing since inflow and outflow compensate. Perhaps it should be "Since oscillatory inflow and outflow components occurring at the same salinity compensate...".

Changed to the suggested phrasing.

5) P3 L2: "TEF profiles computed from numerical model output can be noisy", it should be useful to present some physical reasons of noisy results. Perhaps, within well-mixed space-time domains conversion from depth to salinity coordinates has errors. Other reasons may also exist.

We think the main reason actually lies in the number of salinity bins as the second part of the quoted sentence says. As each data point has probably a different numerical salinity value, although they should have the same value, for increasing $N$ compensating inflow and outflow components do not compensate anymore. For the analytical example the sampling in discrete values did not reproduce the exact salinity values for ebb and flood period, but deviated in some decimal places, which for high enough $N$ created the deviation from the analytical example. We don't convert from depth to salinity coordinates by doing calculations. We use the model's salinities for each cell and only sort the tracer transport of said cell into the corresponding salinity bin. No noise is added by this process.

6) P3 L19: "Obviously, this dividing salinity method only works for classical exchange flows". Please consider if this sentence is needed at this point of introduction.

We reformulated to: "Using the maximum of $Q$ only works for classical two-layer exchange flows."

7) P4 L13: "To visualise why only the dividing salinity method is converging towards the real bulk values" is not clear, "why only" cannot be found. It is not explicitly written by which method Fig. 2a-c have been produced.

Fig. 2a-c are produced with (15) and (16). We added a reference to these equations in the Figure caption.

We deleted the clause "To visualise [...]". The paragraph now starts with: " We created a time series [...]".

8) P4 L24: "The differences of the noises are because of ...", it is not clear the noises of what quantities are considered. It is well known that integrating the discrete curve makes the result smoother, but taking derivatives amplifies the small-scale variations (noise).

We mean that what you said here. We rephrased the sentence to: "The integration process of the discrete $q^c$, see (3), smooths the resulting $Q^c$. "

9) P11 L23-25 : "...TEF bulk values, computed with the sign method. q becomes more noisy with increasing N and causes the sign method to converge towards the absolute exchange values.". It is not clear what the absolute exchange values are and how the convergence is proved.

Please see point 2b).

Please also note the supplement to this comment:
https://www.ocean-sci-discuss.net/os-2018-147/os-2018-147-AC2-supplement.pdf

[Figure]

[Figure]

**Fig. 1.** For the caption, please see the text.

**Supplement:**

**Numerical issues of the Total Exchange Flow (TEF) analysis framework for quantifying estuarine circulation**

Marvin Lorenz1, Knut Klingbeil1, Parker MacCready2, and Hans Burchard1 1Leibniz Institute for Baltic Sea Research Warnemünde, Rostock, Germany 2University of Washington, Seattle, Washington

Correspondence: Marvin Lorenz (marvin.lorenz@io-warnemuende.de)

**Abstract.** For more than a century, estuarine exchange flow has been quantified by means of the Knudsen relations which connect bulk quantities such as inflow and outflow volume fluxes and salinities. These relations are closely linked to estuarine mixing. The recently developed Total Exchange flow (TEF) which uses salinity coordinates to calculate these bulk quantities allows an exact formulation of the Knudsen relations in realistic cases. There are however numerical issues, since the original

5 method does not converge to the TEF bulk values for an increasing number of salinity classes. In the present study, this problem is investigated and the method of *dividing salinities*, described by MacCready et al. (2018), is mathematically introduced. A challenging yet compact analytical scenario for a well-mixed estuarine exchange flow is investigated for both methods, showing the proper convergence of the dividing salinity method. Furthermore, the dividing salinity method is applied to model results of the Baltic Sea to demonstrate the analysis of realistic exchange flows and exchange flows with more than two layers.

**10 1 Introduction**

15

The *Total Exchange Flow* (TEF) analysis framework calculates time-averaged net volume and mass transports between enclosed volumes of the ocean and ambient water masses, sorted by salinity classes. Since oscillatory inflow and outflow components occurring at the same salinity compensate, TEF characterises the net exchange flow with the ambient ocean. Salinity rather than density or temperature is used as a coordinate for calculating estuarine exchange flow, since only the salt budget is entirely controlled by the exchange flow. Therefore, salt is the only conserved quantity. In contrast, temperature and

thus density are additionally affected by the freshwater run-off and the surface heat fluxes.

A first bulk approach based on inflow and outflow salinity and volume transport had been developed and applied to the exchange flow of the Baltic Sea by Knudsen (1900). The theoretical framework based on a continuous salinity space had first been developed by Walin (1977), and later been applied to exchange flow in the Baltic Sea (Walin, 1981). A comparable

20 framework had been applied by Döös and Webb (1994) for quantifying meridional overturning circulation in the Southern Ocean. Both the bulk concept by Knudsen (1900) and the continuous concept by Walin (1977) had been consistently combined by MacCready (2011) who also coined the term *Total Exchange Flow*, TEF. TEF considers a time-averaged transport of a tracer c,  $Q^c$ , through the cross-sectional area  $A(S)A(s \ge S)$ , which has a salinity s above a specific value S.  $Q^c$  is defined as

$$Q^{c}(S) = \left\langle \int \underline{A(S)} \underline{A(s)} c u \mathrm{d}A \right\rangle, \tag{1}$$

where u is the incoming velocity normal to A(S) A(s > S) with the definition that positive u brings water into the estuary and
\$\lambda\$ \lambda\$ denotes temporal averaging. The exchange profile of tracer flux per salinity as functions a function of the salinity is then obtained by differentiating Qc(S) with respect to S:

$$q^{c}(S) = -\frac{\partial Q^{c}(S)}{\partial S},\tag{2}$$

such that  $Q^c$  can be also obtained via integration of  $q^c$  in salinity space,

$$Q^{c}(S) = \int_{S' \ge S} q^{c}(S') \, \mathrm{d}S' = \int_{S}^{S_{\max}} q^{c}(S') \, \mathrm{d}S'.$$
(3)

10 Based on these quantities consistent Knudsen bulk values for inflowing and outflowing salinity  $(s_{in}, s_{out})$ , volume flux  $(Q_{in}^{1} = Q_{in}, Q_{out})$  and salt flux  $(Q_{in}^{s}, Q_{out}^{s})$ , obeying

$$s_{\rm in} = \frac{Q_{\rm in}^s}{Q_{\rm in}}, \quad s_{\rm out} = \frac{Q_{\rm out}^s}{Q_{\rm out}}, \tag{4}$$

can be obtained. MacCready (2011) calculates the inflowing and outflowing bulk fluxes by integrating over positive and negative parts of  $q^c$ :

15
$$Q_{\rm in}^{c,\rm sign} = \int_{S_{\rm min}}^{S_{\rm max}} (q^c)^+ \,\mathrm{d}S, \quad Q_{\rm out}^{c,\rm sign} = \int_{S_{\rm min}}^{S_{\rm max}} (q^c)^- \,\mathrm{d}S,$$
 (5)

where for any function a, the positive part is calculated as  $(a)^+ = \max(a, 0)$  and the negative part is calculated as  $(a)^- = \min(a, 0)$ . In (5),  $S_{\min}$  and  $S_{\max}$ ,  $S_{\min}$  and  $S_{\max}$  are the minimum and maximum salinities. We will call this method of integrating positive and negative contributions separately to obtain  $Q_{in}^c$  and  $Q_{out}^c$  sign method in the following. Bulk salinities are defined as the fractions between the salinity fluxes,  $Q_{in}^s$  and  $Q_{out}^s$ , and the volume fluxes,  $Q_{in} = Q_{in}^1$  and  $Q_{out} = Q_{out}^1$ :

20
$$s_{\text{in}} = \frac{Q_{\text{in}}^s}{Q_{\text{in}}}, \quad s_{\text{out}} = \frac{Q_{\text{out}}^s}{Q_{\text{out}}}.$$

**Recently, Klingbeil et al. (2018a)-**

Recently, Klingbeil et al. (2019) showed the relation between TEF and Thickness Weighted Averaging. The concepts by Knudsen (1900), Walin (1977) and MacCready (2011) were focussed on estuarine systems, which are characterised by distinct volume inflow  $Q_r$  of water masses of (almost) zero salinity. The exchange flow between the estuary and the ocean is described

25 by the Knudsen bulk values, which are volume inflow and outflow of saline water masses, Qin and Qout as well as associated

inflow and outflow salinities, *s*in and *s*out. The Total Exchange Flow provides one consistent calculation method for these bulk values, which for this case describe the net exchange flow. Since there is no clear definition of the Knudsen bulk values, we will call these Total Exchange Flow *TEF bulk values* to distinguish between other bulk values which also fulfill the Knudsen relations, e.g. bulk values computed from a Eulerian version of TEF. The Knudsen relations have been reviewed in detail for

- 5 exchange flow in the Western Baltic Sea by Burchard et al. (2018a). Recently, MacCready et al. (2018) showed how the bulk concept can be used to estimate the volume-integrated average mixing M (defined as the rate of reduction of the net salinity variance due to mixing) in estuaries:  $M \approx s_{in}s_{out}Q_r$ , i.e. the volume-integrated average mixing in an estuary is approximated by the product of inflow and outflow salinity with the estuarine freshwater supply. This mixing estimate by MacCready et al. (2018) approximates the TEF-based exact formulations developed by Burchard et al. (2018b).
- Since the TEF analysis framework is continuous in salinity, a discretisation in salinity space is required when analysing results data from numerical model simulations or field observations. In their Appendix A2, Klingbeil et al. (2018b) Klingbeil et al. (2019) p the remapping of discrete data into bins. As a result, the output of a numerical model consists of a finite number of transport values associated with the same number of discrete salinities. TEF profiles computed from numerical model output can be Comparable to a histogram, the transport data are binned into salinity classes according to their associated salinities. As
- 15 discussed by MacCready et al. (2018), the resulting TEF profiles can become noisy, i.e. sign changes in qc, when the number of discrete salinity classes N is chosen too highas discussed by MacCready et al. (2018). This leads to incorrect TEF bulk values since as described above only the sign is used to distinguish between inflow and outflow. In the limit of N → ∞, meaning each transport valuehas its own salinity class, the bulk values do not converge towards the correct ones, but rather towards absolute values, e.g. for the volume inflow: . For data sets with pairwise disjunct salinities the number of transport values assigned to a single salinity bin decreases with the number of the salinity bins. After exceeding a threshold number of salinity classes, the
- bins will be sufficiently small to hold at most one transport value. In this case  $Q_{in}^{sign}$  is equal to  $Q_{in}^{abs}$ , with

$$\lim_{N \to \infty} Q_{in}^{\rm sign}(N) = Q_{in}^{\rm abs} \neq Q_{\rm in}$$

$$Q_{\rm in}^{\rm abs} = \left\langle \int_{A} u^+ \mathrm{d}A \right\rangle. \tag{6}$$

25 with  $Q_{in}^{\text{sign}}$  being the inflowing volume flux computed with In most practical applications the salinity data are neither constant in space nor time and in the limit of an infinite number of salinity classes  $Q_{in}^{\text{sign}}$  will converge to  $Q_{in}^{\text{abs}}$  which is not the desired result for  $Q_{in}$ .

In order to obtain robust bulk values, which are less sensitive to the number of salinity bins, MacCready et al. (2018) suggested an alternative to the sign method, defined in (5) with c = 1, and

30
$$Q_{\rm in}^{\rm abs} = \left\langle \int_A u^+ \mathrm{d}A \right\rangle.$$

MacCready et al. (2018) suggested a way around this problem by finding. Instead of finding an optimal number of bins (a problem well known for histograms (Knuth, 2006)), they suggested to find a *dividing salinity*  $S_{div}$  which separates the inflow and outflow of a classical two-layer estuary with inflow at high and outflow at low salinity classes, i.e.  $q^c(S_{div}) = 0$  and  $Q^c(S_{div}) = \max(Q^c(S))$ . The bulk values for inflow and outflow are then obtained by integrating:

5
$$Q_{\rm in}^{c,{\rm div}} = \int_{S_{\rm div}}^{S_{\rm max}} q^c \,\mathrm{d}S, \quad Q_{\rm out}^{c,{\rm div}} = \int_{S_{\rm min}}^{S_{\rm div}} q^c \,\mathrm{d}S. \tag{7}$$

It should be noted that analytically and for smooth  $q^c$  with only one zero crossing both methods coincide. We will show in Sect. 2 the different convergence behaviours and will show that the *dividing salinity method* indeed converges towards robust TEF bulk values, e.g. for in inflowing volume flux:

$$\lim_{N \to \infty} Q_{\rm in}^{\rm div}(N) = Q_{\rm in},$$

20

10  $\lim_{N \to \infty} Q_{in}^{\text{div}}(N) = Q_{in}$  where  $Q_{in}^{\text{div}}$  denotes the infowing 
[revised manuscript text omitted]